# The Scalable Solid-State Synthesis of a Ni_5_P_4_/Ni_2_P–FeNi Alloy Encapsulated into a Hierarchical Porous Carbon Framework for Efficient Oxygen Evolution Reactions

**DOI:** 10.3390/nano12111848

**Published:** 2022-05-28

**Authors:** Xiangyun Tian, Peng Yi, Junwei Sun, Caiyun Li, Rongzhan Liu, Jian-Kun Sun

**Affiliations:** 1College of Textiles and Clothing, Qingdao University, Qingdao 266071, China; txy3287425946@163.com (X.T.); yp13792668731@163.com (P.Y.); 17864235865@163.com (C.L.); 2College of Chemistry and Chemical Engineering, Qingdao University, Qingdao 266071, China; sunjunwei121@163.com; 3Collaborative Innovation Center for Eco-Textiles of Shandong Province and the Ministry of Education, Qingdao University, Qingdao 266071, China

**Keywords:** FeNi alloy, Ni_5_P_4_/Ni_2_P heterojunction, solid-state grinding, in situ template, oxygen evolution reaction

## Abstract

The exploration of high-performance and low-cost electrocatalysts towards the oxygen evolution reaction (OER) is essential for large-scale water/seawater splitting. Herein, we develop a strategy involving the in situ generation of a template and pore-former to encapsulate a Ni_5_P_4_/Ni_2_P heterojunction and dispersive FeNi alloy hybrid particles into a three-dimensional hierarchical porous graphitic carbon framework (labeled as Ni_5_P_4_/Ni_2_P–FeNi@C) via a room-temperature solid-state grinding and sodium-carbonate-assisted pyrolysis method. The synergistic effect of the components and the architecture provides a large surface area with a sufficient number of active sites and a hierarchical porous pathway for efficient electron transfer and mass diffusion. Furthermore, a graphitic carbon coating layer restrains the corrosion of alloy particles to boost the long-term durability of the catalyst. Consequently, the Ni_5_P_4_/Ni_2_P–FeNi@C catalyst exhibits extraordinary OER activity with a low overpotential of 242 mV (10 mA cm^−2^), outperforming the commercial RuO_2_ catalyst in 1 M KOH. Meanwhile, a scale-up of the Ni_5_P_4_/Ni_2_P–FeNi@C catalyst created by a ball-milling method displays a similar level of activity to the above grinding method. In 1 M KOH + seawater electrolyte, Ni_5_P_4_/Ni_2_P–FeNi@C also displays excellent stability; it can continuously operate for 160 h with a negligible potential increase of 2 mV. This work may provide a new avenue for facile mass production of an efficient electrocatalyst for water/seawater splitting and diverse other applications.

## 1. Introduction

Developing eco-friendly and sustainable energy technologies is urgent due to the need to address global environmental issues and energy depletion; nevertheless, it remains a challenge [1]. Hydrogen energy is considered to be a promising alternative to conventional fossil fuels [2]. Electrochemical water splitting presents an effective and economical approach to producing clean hydrogen, but the sluggish kinetics of the oxygen evolution reaction (OER) on anodes require a large overpotential to undergo a four-electron transfer process, which severely impedes the overall efficiency of water splitting [3,4]. At present, Ir/Ru-based catalysts are considered the most advanced catalysts for the OER, but their scarcity and high cost hinder the expansion of large-scale industrial applications [5]. Therefore, much effort has been devoted to developing highly active, cost-effective and earth-abundant non-precious-metal electrocatalysts [6]. Recently, a tremendous amount of research has been focused on 3D transition-metal-based non-noble catalysts [7], including transition metal oxides/hydroxides, sulfides, selenides, alloys, phosphates, nitrides, phosphides and so on [8].

Among them, transition metal phosphide (TMP) electrocatalysts have attracted widespread attention due to their tunable structure, unique physicochemical properties and high intrinsic catalytic activity [9]. In particular, the phosphorus and metal sites in TMPs can serve as acceptors for protons and hydrides, respectively, which may enhance the intrinsic OER catalytic activity. Several research studies reported that a TMP heterostructure displayed superior performance compared to a single-phase metal phosphide during the phosphorization process [10]. Meanwhile, the formation of a TMP heterojunction interface can effectively lower the chemisorption free energies of H*/OH* and accelerate the separation of intermediates [11], contributing to a superior OER electrocatalytic performance. For instance, a heterostructure interface made of metallic phosphides usually provides abundant active sites and synergistically promotes the kinetics of proton and electron transport, accelerating the OER process [12]. Ren et al. reported that heterogeneous Ni_2_P–Fe_2_P microsheets on nickel foam produced by a growth-ion exchange and phosphidation method exhibit excellent catalytic activity [13]. Hou et al. reported that Ni/Ni_2_P hetero-nanoparticles on N-doped carbon nanofiber catalysts displayed good OER activity, with an overpotential of 285 mV (10 mA cm^−2^) in a 1 M KOH solution [14]. In addition, TMP catalysts usually exhibit surface reconstruction under OER conditions; this generates amorphous or metal (oxy)hydroxide species that are recognized as the real active sites and are responsible for enhanced levels of activity. For example, hollow nanostructured Ni_5_P_2_/FeP_4_ nanoboxes undergo deep reconstruction to NiOOH/FeOOH nanosheets, which exhibit superior OER activity and stability [15]. Although well-defined heterostructured catalysts usually exhibit superior electrocatalytic activity, TMPs demonstrate thermodynamic instability at high oxidation potentials due to the dissolution of the phosphorus in the electrolyte [16]. Furthermore, TMP catalysts with a well-defined heterostructure usually require tedious synthesis procedures, so feasible fabrication and the scale-up of production remain a challenge for practical applications.

In addition to TMPs, transition metal alloys have recently been studied as promising OER electrocatalysts [17]. A simple mechanical alloying process that involves the physical mixing of diverse elements is the common approach. The multiple components in transition metal alloys can synergistically regulate the electronic structure to promote conductivity and charge transfer [18]. However, bare alloy catalysts suffer from severe erosion and aggregation during electrochemical cycles; in particular, 3D transition metal alloys suffer serious instability in highly oxidative operating conditions and conditions with high levels of alkaline electrolytes, causing a drastic decline in catalytic performance [19]. The coupling/embedding of alloys with carbon-based substances seems an effective approach to enhancing the OER activity and stability [20] since carbon-based layers wrapped around the alloys prevent direct exposure to electrolytes and inhibit the agglomeration of adjacent metal particles [21]. Wei et al. prepared a catalyst of NiFe alloy nanoparticles encapsulated in nitrogen-doped carbon nanofibers (NiFe@NCNFs) using an electrospinning method; this catalyst exhibited enhanced OER activity and durability. Nevertheless, alloy particles are inclined to aggregate to large crystal sizes during the alloying process, which reduces the accessible surface area and number of catalytic sites [22].

Inspired by the above background, we herein develop a method for the facile and scalable synthesis of a Ni_5_P_4_/Ni_2_P heterojunction and FeNi alloy hybrid encapsulated by three-dimensional hierarchical porous carbon (denoted by Ni_5_P_4_/Ni_2_P–FeNi@C) via a room-temperature solid-state grinding and sodium-carbonate-assisted pyrolysis strategy [18]. Unlike the previous works [17,18,19,20,21,22], the sodium-carbonate-assisted pyrolysis strategy can simultaneously induce the in situ generation of a template and pore-former. The process can not only impart a 3D porous nanocrystal-assembled carbon skeleton but also restrain the excessive coalescence of alloy particles and assist in implanting the FeNi alloy into the carbon framework [23]. After phosphorization treatment, the integrated electrocatalyst comprises a Ni_5_P_4_/Ni_2_P heterojunction and FeNi alloy encapsulated into a carbon shell, dispersedly interspersed into the interconnected carbon framework. The components and architectures create synergistic effects between the Ni_5_P_4_/Ni_2_P hetero-interfaces and the FeNi alloy; these effects, as well as the hierarchical porous carbon, regulate the electronic structure and provide a large surface area with a sufficient number of available active sites, contributing to efficient electron transfer and mass diffusion. Consequently, the Ni_5_P_4_/Ni_2_P–FeNi@C catalyst displays superior OER activity compared to commercial RuO_2_, with an overpotential of 242 mV (10 mA cm^−2^) and long-term stability in a 1 M KOH electrolyte solution. Furthermore, the Ni_5_P_4_/Ni_2_P–FeNi@C catalyst also shows promising potential for application in seawater electrolysis, requiring an overpotential of 445 mV to deliver 500 mA cm^−2^ in alkaline natural seawater at 25 °C. Ni_5_P_4_/Ni_2_P–FeNi@C also retains an extraordinary long-term stability; it lasts for 160 h with a negligible potential increase of 2 mV in a 1 M KOH + seawater medium.

## 2. Materials and Methods

### 2.1. Reagents and Chemicals

The reagents used included nickel chloride hexahydrate (NiCl_2_ 6H_2_O), nickel nitrate hexahydrate (Ni(NO_3_)_2_ 6H_2_O), ferric chloride hexahydrate (FeCl_3_ 6H_2_O), ferric nitrate hexahydrate (Fe(NO_3_)_3_ 6H_2_O), anhydrous sodium carbonate (Na_2_CO_3_), sodium bicarbonate (NaHCO_3_), chitosan ((C_6_H_11_NO_4_)_n_), potassium hydroxide (KOH, 98%), absolute ethanol (C_2_H_5_OH) and sodium chloride (NaCl). All of the above were analytical grade (AR) and were purchased from Sinopharm Group. Nafion (C_5_HF_17_O_5_S, 5%) and ruthenium oxide (RuO_2_) were purchased from Shanghai Aladdin Biochemical Technology Co., Ltd. (Shanghai, China).

### 2.2. Preparation of the Ni–FeNi@C and FeNi_3_@AC Catalysts

Appropriate amounts of FeCl_3_·6H_2_O (0.625 mmol), NiCl_2_·6H_2_O (1.876 mmol), Na_2_CO_3_ (4.717 mmol) and chitosan (1 g) were vigorously ground in an agate mortar to a homogenous mixture. Then, these powders were annealed at 700 °C for 2 h in an argon atmosphere. Subsequently, the product was rinsed repeatedly with de-ionized water and ethanol, followed by drying at 60 °C for 2 h. The carbonized sample was named Ni–FeNi@C. The contrast samples were prepared without the addition of sodium carbonate and are denoted as FeNi_3_@AC.

### 2.3. Preparation of the Ni_5_P_4_/Ni_2_P–FeNi@C and Ni_5_P_4_/Ni_2_P–Fe–FeNi_3_@AC Catalysts

For a typical preparation of Ni_5_P_4_/Ni_2_P–FeNi@C, 1 g NaH_2_PO_2_ on a quartz boat was placed on the upstream side of the tube furnace, while 100 mg of Ni–FeNi@C was placed on the downstream side. Then, the furnace was kept at 350 °C for 2 h at 5 °C min^−1^ in an argon atmosphere. The final black products were denoted as Ni_5_P_4_/Ni_2_P–FeNi@C. When the precursor was FeNi_3_@AC, the phosphating sample was labeled as Ni_5_P_4_/Ni_2_P–Fe–FeNi_3_@AC.

### 2.4. Characterization

Morphological features were evaluated by SEM (JSM-7001F, JEOL, Tokyo, Japan) and TEM (Tecnai G2F30, Hillsboro, OR, USA). XPS was performed on an ESCALAB 250Xi X-ray photoelectron spectrometer (Thermo Scientific, Waltham, MA, USA). XRD patterns were tested on DX2700 equipment (Dandong, China). The Nitrogen adsorption–desorption measurement was performed by a physical adsorption apparatus (ASAP 2020, micromeritics). Raman spectra analyses were performed on LabRAM Aramis (Raman, HORIBA, Ltd., Kyoto, Japan) using a 532 nm excitation laser.

### 2.5. Electrochemical Measurements

All electrochemical tests were carried out using an electrochemical workstation (VSP-300, BioLogic, Seyssinet-Pariset, France). An EIS test was performed at an amplitude of 5 mV and at frequencies ranging from 10^6^ to 0.01 Hz. All potentials against Hg/HgO (E_Hg/HgO_) were converted into the reversible hydrogen electrode (RHE) using the following equation:E_RHE_ = E_Hg/HgO_ + 0.0592 × pH + 0.098(1)

Catalyst ink was prepared by dispersing 5 mg catalyst in a 500 µL solution containing 490 µL of ethanol/water (the volume ratio was 1:1) and 10 µL of 5 wt% Nafion. The working areas of GCE and carbon cloth were 0.07065 and 0.25 cm^−2^, respectively. Quantities of 5 μL and 17.7 μL catalyst ink were dropped on the GCE and carbon cloth substrates, respectively.

## 3. Results and Discussion

### 3.1. Schematic Diagram of the Synthesis Process

As schematically illustrated in Figure 1, the catalysts were prepared via a solid grinding method followed by a carbonization pyrolysis and phosphorization process. Briefly, a homogenous solid mixture of hydrated metal chloride (including FeCl_3_·6H_2_O and NiCl_2_·6H_2_O), Na_2_CO_3_ and chitosan was obtained by vigorously grinding in an agate mortar. Here, chitosan serves as a carbon source. This product was labeled as Ni–FeNi@C, and the product without the addition of Na_2_CO_3_ was labeled as FeNi_3_@AC. Then, the mixture was calcined at 700 °C in an argon atmosphere. Subsequently, the pyrolysis product was rinsed repeatedly with de-ionized water and ethanol. Finally, the pyrolysis sample underwent a phosphorization process with NaH_2_PO_2_ as its phosphorous source. The phosphatized samples from the initial mixture with or without Na_2_CO_3_ were labeled as Ni_5_P_4_/Ni_2_P–FeNi@C or Ni_5_P_4_/Ni_2_P–Fe–FeNi_3_@AC, respectively. More experimental details were presented in the Experimental Section and Appendix A.

### 3.2. Structural Analysis

X-ray diffraction (XRD) measurement was employed to study the composition of the samples obtained by Na_2_CO_3_-assisted pyrolysis. As the XRD patterns of the pyrolysis products show in Figure 2a, the diffraction peak at 26.3° is well indexed to the (002) plane of graphitic carbon (JCPDS No.41-1487); the diffraction peaks at 44.3°, 51.6° and 76° are ascribed to the (111), (200) and (220) planes, respectively, of metal Ni (JCPDS No.97-007-6667). In addition, the peaks at 43.5°, 50.7° and 74.5° are ascribed to the (111), (200) and (220) planes, respectively, of FeNi alloys (JCPDS No.97-063-2933). The results indicate that the metal Ni and FeNi hybrid alloys, as well as the graphitic carbon matrix (denoted by Ni–FeNi@C), are obtained by the Na_2_CO_3_-assisted pyrolysis. The XRD pattern in Figure 2b reveals the appearance of several new diffraction peaks, but the original peaks of graphitic carbon and the metal NiFe alloy, as well as the new diffraction peaks, are well indexed to Ni_5_P_4_ (JCPDS No.89-2588) and Ni_2_P (JCPDS No.74-1385). These results suggest the metal Ni in the Ni–FeNi@C sample converts into a Ni_5_P_4_ and Ni_2_P hybrid, while the FeNi alloy and graphitic carbon are well preserved; the result is labeled as Ni_5_P_4_/Ni_2_P–FeNi@C.

For comparison, a similar pyrolysis and phosphorization route was repeated based on the same solid-state mixture except without Na_2_CO_3_. The XRD pattern in Figure 2c shows the peaks that are assigned to the FeNi_3_ alloy (JCPDS No.65-3244); the broad bulge at 20–30° is ascribed to the amorphous carbon, indicating that the sample obtained by carbonization pyrolysis without the addition of the Na_2_CO_3_ precursor is composed of the FeNi_3_ alloy and amorphous carbon (denoted by FeNi_3_@AC). Meanwhile, the XRD pattern of the phosphating product in Figure 2d detects Ni_5_P_4_ (JCPDS No.89-2588), Ni_2_P (JCPDS No.74-1385), the FeNi_3_ alloy (JCPDS No.65-3244) and even metal Fe (JCPDS No.97-004-4862), but not amorphous carbon, indicating that FeNi_3_@AC undergoes Fe-leaching together with the phosphorization of Ni metal during the phosphorization process; the result is labeled as Ni_5_P_4_/Ni_2_P–Fe–FeNi_3_@AC.

These controlled experiments reveal that Na_2_CO_3_ plays a vital role in establishing the composition and phase state of the carbonization pyrolysis product, and that it has an effect on the phosphatized sample [24]. To determine the effect of Na_2_CO_3_, the initial solid-state grinding mixture was evaluated by XRD. As shown in Appendix A, the diffraction peaks of NaCl (JCPDS No.99-0059) and NaHCO_3_ (JCPDS No.15-0700) are detected in the ground mixture that includes the Na_2_CO_3_ precursor, while only the original metal chloride diffraction peaks are found in the mixture without the addition of Na_2_CO_3_ (Appendix A), indicating that NaCl and NaHCO_3_ are formed in situ during the physical grinding of Na_2_CO_3_ and hydrated metal chloride. The sodium ion (Na^+^) captures chloride ions (Cl^−^) in metal salt to form NaCl; meanwhile, the excess Na_2_CO_3_ reacts with H_2_O molecules from hydrated metal chloride to gradually generate NaHCO_3_. Thus, the difference observed for Na_2_CO_3_-assisted pyrolysis can be ascribed to the synergetic effect of NaCl and NaHCO_3_. Therefore, it can be concluded that NaCl and NaHCO_3_ command the Ni–Fe-based alloy phase and the carbonization of chiston. NaCl and NaHCO_3_ impel the precursors to form the stable FeNi alloy with the residual Ni metal and well aligned graphitic carbon [25]. In contrast, the metal chloride and chitosan precursor is inclined to generate the FeNi_3_ alloy and amorphous carbon without the addition of Na_2_CO_3_. In the subsequent phosphorization process, the residual Ni metal in the Ni–FeNi@C sample converts into Ni_5_P_4_/Ni_2_P while the FeNi alloy is well preserved, whereas the FeNi_3_ alloy in the FeNi_3_@AC sample results in Ni_5_P_4_/Ni_2_P and the occurrence of Fe-leaching [26].

The morphology and structure of the as-fabricated samples were investigated by scanning electron microscope (SEM) and transmission electron microscope (TEM). A typical SEM image of FeNi_3_@AC (Figure 3a) exhibits vast agglomerated alloy particles of micron size, which are irregularly dispersed in the caked carbon matrix. A zoomed-in image (Figure 3b) displays these alloy particles are, on the whole, suspended and not in close contact with the carbon substrate. After the phosphorization treatment, the particles in Ni_5_P_4_/Ni_2_P–Fe–FeNi_3_@AC (Figure 3c) explode, and a magnified image in Figure 3d shows that the smooth surface becomes rough and resembles the texture of pinecones. However, the samples created with the Na_2_CO_3_ precursor, Ni–FeNi@C, as shown in Figure 3e, present a porous granular morphology with nanoparticle clusters. A magnified image (Figure 3f) displays a mass of nanoparticle-assembled sheets interconnected to build a hierarchical porous carbon framework; metal/alloy particles of nanometer size are interspersed in the carbon framework. A detailed observation of the section of the image marked by a white gridline (Figure 3f) reveals that the carbon layer may wrap around the particle. Owing to the thermal decomposition of NaHCO_3_ as a pore-forming agent, the carbon framework becomes loose and porous, which enhances the specific surface area and accelerates electron transfer. Compared to Ni–FeNi@C, the Ni_5_P_4_/Ni_2_P–FeNi@C sample contains more dense particles but retains the porous carbon skeleton (Figure 3g,h).

In order to further reveal the role of Na_2_CO_3_, three groups of controlled trials were carried out. Briefly, sole NaCl, sole NaHCO_3_ and a mixture of NaCl and NaHCO_3_ were ground with hydrated metal nitrate (including Fe(NO_3_)_3_·6H_2_O and Ni(NO_3_)_2_·6H_2_O) and chitosan. These alternative sources of hydrated metal nitrate, as opposed to hydrated metal chloride, avoid the possible formation of NaCl. The process followed was consistent with the previous preparation conditions (See the Experimental Section). The final pyrolysis products were labeled as S-NaCl, S-NaHCO_3_ and M-NaCl/NaHCO_3_, respectively. As shown in Appendix A, the sole NaCl-assisted pyrolysis product is composed of a particle-assembled sheet, and the size of the particles is less than 100 nm (Appendix A). In contrast, the SEM image of S-NaHCO_3_ in Appendix A displays a loose and porous morphology, and metal particles of a large size (about 500 nm) are embedded in the porous framework. Therefore, it can be deduced that NaCl, as a template, leads to the formation of dispersive metal particles with a smaller size [26], whereas NaHCO_3_ plays the role of pore-former. More interestingly, M-NaCl/NaHCO_3_, as shown in Appendix A, shows a similar morphology to that of Ni–FeNi@C, indicating that Na_2_CO_3_ plays a synergetic role with effects on NaCl and NaHCO_3_. The XRD test result in Appendix A shows that the products of both the sole NaCl-assisted pyrolysis and the pyrolysis performed with the mixture of NaCl and NaHCO_3_ contain Ni metal, the FeNi alloy and graphitic carbon. In contrast, the sole NaHCO_3_-assisted pyrolysis product contains only the FeNi_3_ alloy and graphitic carbon. Therefore, it can be deduced that NaCl, as a template, impels the formation of the FeNi alloy phase. Based the above results, it can also be confirmed that Na_2_CO_3_ has synergetic effects as a template and pore-former. The in situ generated NaCl, when used as a template, conducts the formation of the FeNi alloy phase and inhibits the coarsening of the alloy particles, while NaHCO_3,_ as a pore-former, imparts a loose and porous morphology that enhances the specific surface area.

TEM images show that the size of the particles in Ni–FeNi@C (Appendix A) is far smaller than that of the particles in FeNi_3_@AC (Appendix A), which is in accordance with the SEM results. It is reasonable to deduce that NaCl generated from the addition of Na_2_CO_3_ may act as a template to affect the formation of the alloy phase and inhibit the coarsening of the alloy particles [27]. Moreover, NaCl also affects the carbonization pyrolysis of chitosan, so the carbon substrate in FeNi_3_@AC is amorphous, and FeNi_3_ alloy particles are overlaid on the amorphous carbon substrate (Appendix A). However, the carbon layer in Ni–FeNi@C is graphitic and carbonaceous (Figure 4a); this is also verified by the XRD results (Figure 2a). High-resolution TEM (HRTEM) images of the Ni–FeNi@C sample, shown in Figure 4b,c, reveal that inter-planer spacings of 0.179 and 0.203 nm are assigned to the (200) plane of the FeNi alloy and the (111) plane of metal Ni, respectively, suggesting that both the Ni particles and the FeNi alloy particles are embedded in carbon layers [28]. After phosphorization, the architecture of Ni_5_P_4_/Ni_2_P–FeNi@C still maintains its initial microstructure, with the carbon coating wrapped around particles (Figure 4d,e). The HRTEM image in Figure 4f demonstrates that an inter-planar spacing of 0.207 nm is assigned to the (111) plane of the FeNi alloy. Furthermore, the coherent lattice fringe indicates that the crystal structure of the FeNi alloy is well preserved, without any surficial phosphatized reaction during the phosphorization procedure. In contrast, the metal Ni particles in Ni–FeNi@C sample undergo a conversion to Ni_5_P_4_/Ni_2_P heterojunction particles, which is evident in Figure 4g. Spacings of 0.248 and 0.221 nm can be assigned to the Ni_5_P_4_ (104) and Ni_2_P (111) planes, respectively. As seen in the phase boundary marked by the dashed line, Ni_5_P_4_ and Ni_2_P constitute well-defined hetero-interfaces, exposing more active sites for enhanced catalytic activity. The selective region electron diffraction (SAED) pattern in Figure 4h presents the (103) plane of Ni_5_P_4_, the (302) plane of Ni_2_P and the (111) plane of the FeNi alloy. The scanning TEM (STEM) image and the energy dispersive X-ray spectroscopy (EDS) elemental mapping images (Figure 4i) also show the elements Ni, Fe, P and C uniformly distributed throughout the selected region. In addition, the corresponding EDS spectra (Appendix A) and elemental contents (Appendix A) also verify the elemental species and amounts in the Ni_5_P_4_/Ni_2_P–FeNi@C sample.

The specific surface area and pore structure were also investigated by the nitrogen adsorption/desorption isotherm method. In Figure 5a, all samples conform to the type-IV adsorption isotherm. When P/P_0_ ranges from 0.5 to 1.0, the four samples display a hysteresis curve along with a hysteresis loop, suggesting that all four samples have a certain mesoporous structure [29]. In addition, the BET surface areas of FeNi_3_@AC and Ni–FeNi@C are estimated to be 143.32 and 182.12 m^2^ g^−1^, respectively. The higher surface area of the Ni–FeNi@C sample is mainly caused by the pyrolysis of NaHCO_3_; furthermore, NaHCO_3_ affects the carbonization process of chitosan to increase the affinity for alloy particles. Thus, the pyrolytic carbon layer encapsulates the alloy to assist in the condensation process, followed by more dispersive particles and pore architecture, which are also verified by SEM. Meanwhile, the more porous structure facilitates the penetration of PH_3_ gas during the phosphating process to generate more phosphide-based active sites. For the corresponding phosphatized product, the BET surface areas of Ni_5_P_4_/Ni_2_P–Fe–FeNi_3_@AC and Ni_5_P_4_/Ni_2_P–FeNi@C are 12.12 and 82.13 m^2^ g^−1^, respectively. Compared with the carbonized samples, the surface area of the phosphatized product decreases. This occurs because PH_3_ gas from the NaH_2_PO_4_ precursor can penetrate into porous structures and react with FeNi_3_@AC and Ni–FeNi@C, which converts Ni to Ni_5_P_4_/Ni_2_P. The phosphating process can not only cause the expansion of suspended particles to consume the pore volume (Figure 3c,d) but also lead to the collapse and agglomeration of the pore architecture (Figure 3g,h), which, in turn, leads to a reduced surface area. The corresponding average pore sizes of Ni_5_P_4_/Ni_2_P–Fe–FeNi_3_@AC and Ni_5_P_4_/Ni_2_P–FeNi@C are 20.48 and 8.23 nm, respectively, in the Barrett–Joyner–Halenda (BJH) pore size distribution curves (Appendix A), but Ni_5_P_4_/Ni_2_P–FeNi@C displays a more uniformly porous nature. Although the phosphating process reduces the surface area, which may be ascribed to the fact that the expansion consumes the pore structure, Ni_5_P_4_/Ni_2_P–FeNi@C still presents a decent surface area and pore structure.

The Raman spectrum was used to illustrate the carbon structure of the Ni_5_P_4_/Ni_2_P–FeNi@C and Ni_5_P_4_/Ni_2_P–Fe–FeNi_3_@AC samples. As shown in Figure 5b, the D band (~1350 cm^−1^) and G band (~1576 cm^−1^) originate from disorder-induced carbon and graphitized sp^2^ hybrid carbon, respectively. The intensity ratio of the D to G bands (I_D_/I_G_) usually reveals the graphitic degree of carbon materials [13]. A larger I_D_/I_G_ value indicates a lower degree of graphitization [29]. Ni_5_P_4_/Ni_2_P–FeNi@C exhibits a lower I_D_/I_G_ value of 1.14 compared to Ni_5_P_4_/Ni_2_P–Fe–FeNi_3_@AC with a value of 1.36. This difference is due to the formation of NaCl, which improves the carbonization pyrolysis degree of carbonaceous materials [23]; this aligns well with the XRD results. As a result, Ni_5_P_4_/Ni_2_P–FeNi@C possesses a superior electrical transfer capacity and a graphitic carbon protective layer, contributing to superior catalytic activity and long-term durability.

### 3.3. Electrochemical Measurements

The OER performances of Ni_5_P_4_/Ni_2_P-FeNi@C electrocatalyst were evaluated by a glassy carbon electrode configuration in O_2_-saturated 1.0 M KOH electrolyte. To evaluate the effect of different precursor amounts and reaction conditions on the catalytic properties, the 85% iR-corrected linear sweep voltammetry curves (LSV) of Ni_5_P_4_/Ni_2_P-FeNi@C catalyst with different synthesis parameters. including the elemental ratio of nickel and iron precursors, the total amount of metals, the amount of Na_2_CO_3_ and the temperature of carbonization pyrolysis, were firstly investigated in 1.0 M KOH solution. As display in Appendix A, Ni_5_P_4_/Ni_2_P-FeNi@C catalyst prepared based on the ratio of Ni:Fe in the proportion 3:1, the pyrolysis temperature of 700 °C, the total amount of metals with 15% and 0.5 g Na_2_CO_3_, displays the excellent OER catalytic activity, thus the Ni_5_P_4_/Ni_2_P-FeNi@C catalyst with the above optimum synthesis conditions is used for the following studies.

We compared the OER performance of the sample Ni_5_P_4_/Ni_2_P–FeNi@C with FeNi_3_@AC, Ni–FeNi@C, Ni_5_P_4_/Ni_2_P–Fe–FeNi_3_@AC and commercial RuO_2_ catalysts. As shown in Figure 6a, Ni_5_P_4_/Ni_2_P–FeNi@C loaded on the glass carbon electrode (GCE) exhibited excellent OER catalytic activity, requiring the lowest observed overpotential of 242 mV to achieve 10 mA cm^−2^ in 1 M KOH (with 85% iR-correction), which significantly surpasses the performance of Ni_5_P_4_/Ni_2_P–Fe–FeNi_3_@AC (294 mV), Ni–FeNi@C (316 mV), FeNi_3_ @AC (345 mV) and RuO_2_ (272 mV). All overpotential results were obtained with 85% iR-correction unless otherwise specified. Figure 6b shows the detailed comparative overpotential values required to achieve 10 mA cm^−2^. Interestingly, the Ni_5_P_4_/Ni_2_P–FeNi@C catalyst obtained by the ball-milling method displayed a similar level of activity to the one created with the grinding method; as shown in Appendix A, the overpotential for the ball-milling sample was only 248 mV vs. 242 mV at 10 mA cm^−2^, which makes large-scale production possible.

To further evaluate the catalytic properties, the OER dynamics were evaluated by Tafel slope. Ni_5_P_4_/Ni_2_P–FeNi@C exhibited the lowest Tafel slope of 46 mV dec^−1^, as shown in Figure 6c, indicating that Ni_5_P_4_/Ni_2_P–FeNi@C has the most rapid kinetics for the OER process [30]. These results may be caused by the in situ formation of NaCl and NaHCO_3_. Specifically, NaCl as a template establishes the composition and size of the alloy/metal particles as well as the carbon matrix; meanwhile, NaHCO_3_ increases the porous architecture during the carbonization pyrolysis process. Thus, the phosphatized catalyst integrates the merits of having a sufficient number of active sites, thanks to the Ni_5_P_4_/Ni_2_P heterojunction, with an efficient electron/mass transfer capacity thanks to the hierarchically interlaced porous carbon scaffold and the alloy particles [24,31].

The electrochemical impedance spectroscopy (EIS) measurements demonstrate that Ni_5_P_4_/Ni_2_P–FeNi@C displays a smaller charge transfer resistance than the other as-prepared samples at 1.480 V vs. RHE. As seen in the fitting equivalent circuit model in Figure 6d, Ni_5_P_4_/Ni_2_P–FeNi@C has an extremely small R_ct_ of 19.46 Ω, which is much lower than the values for Ni_5_P_4_/Ni_2_P–Fe–FeNi_3_@AC (66.42 Ω), Ni–FeNi@C (149.6 Ω) and FeNi_3_@AC (209.7 Ω). The results suggest that Ni_5_P_4_/Ni_2_P–FeNi@C contributes to a rapid charge transfer, accelerating OER dynamics. The electrochemically active surface area (ECSA) is normalized to obtain the electric double-layer capacitance (C_dl_), which can reflect an important parameter of the electrochemical reaction kinetics and show the intrinsic activity of the catalyst. The charging current obtained at different scan rates is I_C_, and its relationship with the scan rate V and the double-layer capacitance C_dl_ is I_C_ = V × C_dl_. The electrochemically active area of the catalyst is calculated according to the following equation: ECSA = C_dl_/C_S_ (where C_S_ is the specific capacitance of the corresponding smooth surface sample under the same conditions). Therefore, the double-layer capacitance (C_dl_), determined by cyclic voltammograms versus scan rates (Appendix A), is calculated to evaluate the ECSA. As shown in Figure 6e, Ni_5_P_4_/Ni_2_P–FeNi@C presents a larger C_dl_ value (8.01 mF cm^−2^) than Ni_5_P_4_/Ni_2_P–Fe–FeNi_3_ @AC (4.54 mF cm^−2^), demonstrating an enhanced ECSA with more active sites, which may benefit from the hierarchical porous architecture caused by NaHCO_3_. Impressively, the OER catalytic activity of Ni_5_P_4_/Ni_2_P–FeNi@C in this work is comparable to and may even outperform recently reported alloy- and phosphide-composited catalysts (Figure 6f and Table 1) [14,19,22,29,30,32,33,34,35].

Additionally, Ni_5_P_4_/Ni_2_P–FeNi@C as an OER catalyst demonstrates superior catalytic stability in a 1 M KOH electrolyte. As shown in Figure 6g, the observed potential of the Ni_5_P_4_/Ni_2_P–FeNi@C catalyst can maintain an approximate constant at 20 mA cm^−2^, and the potential only increased 32 mV after 140 h of OER catalysis. For comparison, the observed potential increased 57 mV for Ni_5_P_4_/Ni_2_P–Fe–FeNi_3_@AC, without the graphitic carbon coating layer, after only a 40 h catalysis test; it increased 51 mV for the Ni_5_P_4_/Ni_2_P@C catalyst without alloy particles after a 100 h test. Both an SEM image (Appendix A) and a low-magnification TEM image (Appendix A) show that the Ni_5_P_4_/Ni_2_P–FeNi@C catalyst, even after a long-term test, still maintained its initial microstructure, indicating good structural stability. Meanwhile, HRTEM images in Appendix A suggest that the FeNi alloys are stable and can maintain their initial microstructures well due to the protection of carbon coating. We deduce that the outstanding stability of the Ni_5_P_4_/Ni_2_P–FeNi@C catalyst may be ascribed to the stable FeNi alloy phase and the graphitic carbon coating layer. The in situ sodium chloride serves as a template and manipulates the formation of the alloy phase, enhancing the crystallinity of the carbon materials during the pyrolysis process [36]. Consequently, the alloy particles, wrapped tightly by the graphitic carbon coating layer, are interspersed throughout the hierarchical porous carbon skeleton, which buffers against the harsh electrolysis environment and restrains the structural collapse, contributing to long-term stability.

### 3.4. XPS Survey Spectrum of Ni_5_P_4_/Ni_2_P–FeNi@C before and after OER Reaction

To obtain further insights into the highly active Ni_5_P_4_/Ni_2_P–FeNi@C catalyst, XRD and X-ray photoelectron spectroscopy (XPS) measurements were used to probe the differences in composition and surface valance state before and after the OER catalyst underwent a 140 h stability test at 20 mA cm^−2^. As the XRD pattern of the post-OER catalyst shows in Appendix A, the peaks at 43.5° and 50.7° are assigned to the (111) and (200) planes, respectively, of the FeNi alloys (JCPDS No.97-063-2933), and the diffraction peaks at 40.7°, 46.4° and 48.0° are well indexed for NiOOH (JCPDS No.97-016-5961). Due to the low amount of catalyst on the carbon cloth basement, the intensity of the diffraction peaks is weak, and several signals are even undetected. However, the above XRD result confirms the existence of NiOOH and the FeNi alloy after OER stability. The XPS survey spectra of the Ni_5_P_4_/Ni_2_P–FeNi@C catalyst before and after OER verifies the coexistence of the elements Ni, Fe, O, C and P (Appendix A), which corresponds well with the EDX data. The strong F signal detected on the post-OER catalyst is from the Nafion binder, which is used to adhere catalyst power to the carbon cloth basement.

In the high-resolution P 2p spectra (Figure 7a), the initial spectrum can be deconvolved into three peaks; the fitted peaks at 130.6 and 129.8 eV are assignable to P 2p_1/2_ and P 2p_3/2_, respectively, which are ascribed to P^δ−^ in Ni_5_P_4_ and Ni_2_P [12]. The peak at 134.5 eV is attributable to the P–O bond [13]. After the long-term OER test, the peaks of the M–P bond disappear, and the single peak at 133.8 eV corresponds to P–O. This phenomenon may be caused by massive P leaching and the rearrangement and oxidation of metal phosphide (Ni_5_P_4_-Ni_2_P) after the OER reaction [37]. For the spectrum of O 1s in Figure 7b, the initial peaks at 531.8 and 533.5 eV are assigned to O–H and C–O/C = O, respectively [38]. After the OER test, the emerging peaks at 532.7 and 535.4 eV are attributable to Ni–OOH and residual water molecules, respectively [15].

For Ni 2p (Figure 7c), the initial peaks at 855.6 and 872.9 eV correspond to Ni^2+^ 2p_3/2_ and Ni^2+^ 2p_1/2_, respectively [39], and the peaks located at 856.9 and 874.2 eV are ascribed to Ni^3+^ 2p_3/2_ and Ni^3+^ 2p_1/2_, respectively, which originate from oxidized and phosphatized nickel, respectively. The two satellite peaks (marked by Sat.) are at 861.7 and 879.2 eV; meanwhile, the two fitted peaks at 852.5 (2p_1/2_) and 869.8 eV (2p_3/2_) are ascribed to Ni^0^ that originate from the FeNi alloy [33]. After the OER reaction, the fitted peaks of Ni^2+^ 2p_3/2_ (854.6 eV), Ni^3+^ 2p_3/2_ (856.5 eV) and Ni^0^ 2p_3/2_ (852.2 eV) are still detected, whereas the intensity ratio of Ni^3+^ rapidly increases, indicating the formation of NiOOH species [14]. In addition, the broad peaks at around 860.6 and 878.9 eV can be attributed to an Auger peak of F [38].

The XPS spectrum of Fe 2p (Figure 7d) can be deconvolved into a pair consisting of Fe 2p_3/2_ and Fe 2p_1/2_. Before OER, the peaks at 706.7 and 720.3 eV are attributed to the metallic state of iron (Fe^0^). The two fitted peaks at 710.2 and 723.8 eV correspond to Fe 2p_1/2_ and Fe 2p_3/2_ of Fe^2+^, respectively. In addition, the two peaks at 712.8 and 726.4 eV are assigned to Fe^3+^ from the oxidated iron. The peaks at 716.1 and 730.7 eV can be classified as satellite peaks. After the OER test, the fitted characteristic peaks in Figure 7d confirm that the Fe^0^, Fe^2+^ and Fe^3+^ species still exist, further confirming the existence of the zero-valent FeNi alloy.

All the above results clearly show that the Ni_5_P_4_/Ni_2_P in Ni_5_P_4_/Ni_2_P–FeNi@C undergoes a surface reconstruction to generate a NiOOH active layer accompanied by P leaching during the OER reaction, while the FeNi alloy is well preserved during the OER process. Therefore, the NiOOH and FeNi alloys synergistically serve as the real catalytic active sites. Furthermore, the FeNi alloy contributes ultra-high stability. The remarkable catalytic performance of the FeNi alloy can be attributed to it being wrapped in the graphitic carbon coating layer as well as its distribution throughout the carbon skeleton, thus restraining the corrosion of the alloy particles in the electrolyte and increasing the electron/mass transport capacity.

### 3.5. Electrolyzed Seawater Applications

Considering its excellent OER activity and long-term stability in a 1 M KOH electrolyte, we further investigated the catalytic activity of the Ni_5_P_4_/Ni_2_P–FeNi@C catalyst in alkaline simulated seawater and alkaline natural seawater electrolytes. As shown in Figure 8a,b, the Ni_5_P_4_/Ni_2_P–FeNi@C catalyst displayed outstanding OER catalytic activity in the above two electrolytes. In a 1 M KOH + 0.5 M NaCl medium, the Ni_5_P_4_/Ni_2_P–FeNi@C catalyst loaded on carbon cloth required overpotentials of 319 and 376 mV to afford 100 and 500 mA cm^−2^, respectively (with 85% iR-correction), which is close to its performance in 1 M KOH. In alkaline natural seawater from the Yellow Sea, the deleterious effects of the cation/anion ions, bacteria, micro-organisms and even particulate matter in natural seawater resulted in a certain decline in OER activity. Nevertheless, the Ni_5_P_4_/Ni_2_P–FeNi@C catalyst in 1 M KOH + seawater electrolyte still yielded 100 and 500 mA cm^−2^ current densities, requiring 382 and 445 mV, respectively (with 85% iR-correction); these results are below the 480 mV threshold for the chloride oxidation reaction, and thus the Ni_5_P_4_/Ni_2_P–FeNi@C catalyst showed a thermodynamic OER advantage over the undesired chloride oxidation reaction at a current density of less than 500 mA cm^−2^. Furthermore, the Ni_5_P_4_/Ni_2_P–FeNi@C catalyst displayed remarkable stability in the alkaline natural seawater electrolyte. As the chronopotentiometry curve shows in Figure 8c, the long-term OER process was able to operate for 160 h with a negligible potential increase of 2 mV at a 20 mA cm^−2^ current density in the alkaline natural seawater electrolyte, surpassing the overwhelming majority of reported seawater catalysts.

Finally, the ultra-high OER activity and durability of Ni_5_P_4_/Ni_2_P–FeNi@C can be attributed to the following causes: (1) the in situ formation of NaCl and NaHCO_3_ synergistically regulate the pyrolysis process of alkali metal salts and chitosan carbonaceous material [40]. As a result, the phosphatized product of Ni_5_P_4_/Ni_2_P–FeNi@C encapsulates the Ni_5_P_4_/Ni_2_P heterojunction and FeNi alloy hybrid into a 3D hierarchical porous graphitic carbon framework [41], which not only provides more active sites and facilitates electron/mass transfer [42], but also supplies a graphitic carbon protective layer to reduce the corrosion of alloy particles [43]. (2) Both the NiOOH active layer reconstructed on the surface of Ni_5_P_4_/Ni_2_P and the original FeNi alloy synergistically contribute to the real catalytic active sites. Additionally, the well-preserved and dispersive FeNi alloy particles supply ultra-high durability and electron transfer capacity [44].

## 4. Conclusions

In summary, we developed a facile and scalable strategy for the in situ generation of a template and pore-former using a room-temperature solid-state grinding and sodium-carbonate-assisted pyrolysis method. NaCl, as a template, manipulates the stable FeNi alloy phase and inhibits the excessive coalescence of alloy particles, while NaHCO_3_, as a pore-former, establishes a hierarchically porous carbon framework. After phosphorization treatment, the Ni_5_P_4_/Ni_2_P–FeNi@C catalyst integrates the merits of the Ni_5_P_4_/Ni_2_P heterojunction, the stable FeNi alloy phase and the hierarchical porous nanocrystal-assembled carbon skeleton, which combine to impart abundant active sites, an efficient electron/mass transfer ability, and durable corrosion resistance. Consequently, the Ni_5_P_4_/Ni_2_P–FeNi@C catalyst exhibits an ultra-low overpotential of 242 mV and a low Tafel slope of 46 mV dec^−1^ in 1 M KOH, which outperforms the commercial RuO_2_ (272 mV and 65 mV dec^−1^). Both the NiOOH reconstructed on the surface of Ni_5_P_4_/Ni_2_P and the original FeNi alloy synergistically act as the real catalytic active sites. The FeNi alloy is wrapped by the graphitic carbon layer and distributed throughout the carbon skeleton, which contributes ultra-high stability. The Ni_5_P_4_/Ni_2_P–FeNi@C catalyst also displays excellent OER activity in alkaline natural seawater, requiring a low overpotential of 382 mV to deliver 100 mA cm^−2^. More remarkably, the Ni_5_P_4_/Ni_2_P–FeNi@C catalyst exhibits extraordinary long-term stability and can operate for 160 h with a negligible potential increase of 2 mV in an alkaline natural seawater electrolyte. This work may provide a new avenue for high-performance, low-cost and large-scale OER catalysts by integrating a transition metal phosphide heterojunction and metal alloys.

## Figures and Tables

**Figure 1 nanomaterials-12-01848-f001:**
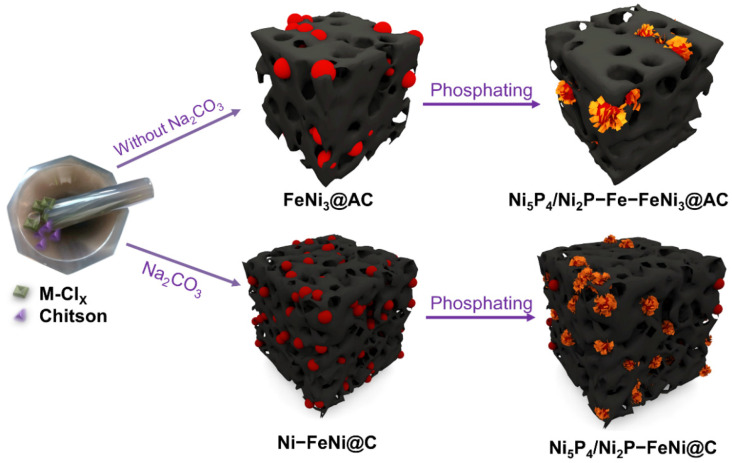
Schematic illustration of the synthesis process for Ni_5_P_4_/Ni_2_P–Fe–FeNi_3_@AC catalysts and Ni_5_P_4_/Ni_2_P–FeNi@C.

**Figure 2 nanomaterials-12-01848-f002:**
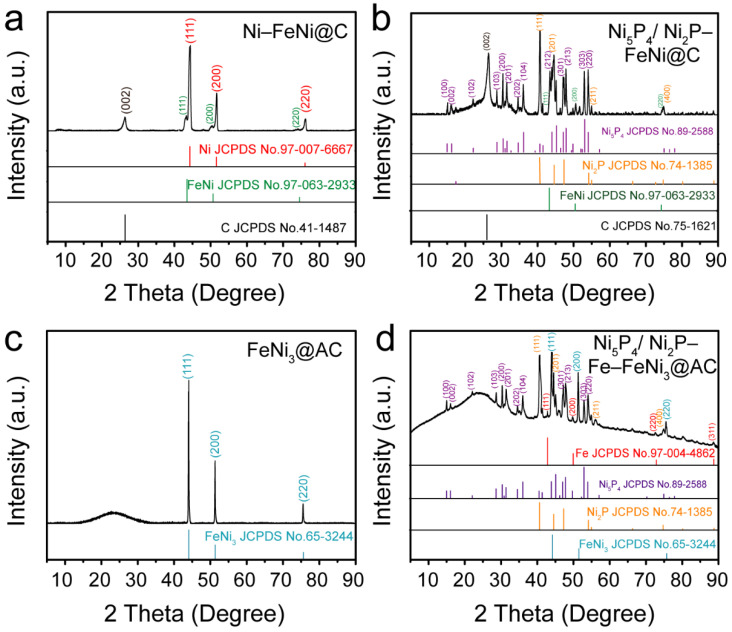
XRD patterns of (**a**) Ni–FeNi@C, (**b**) Ni_5_P_4_/Ni_2_P–FeNi@C, (**c**) FeNi_3_@AC and (**d**) Ni_5_P_4_/Ni_2_P–Fe–FeNi_3_@AC samples.

**Figure 3 nanomaterials-12-01848-f003:**
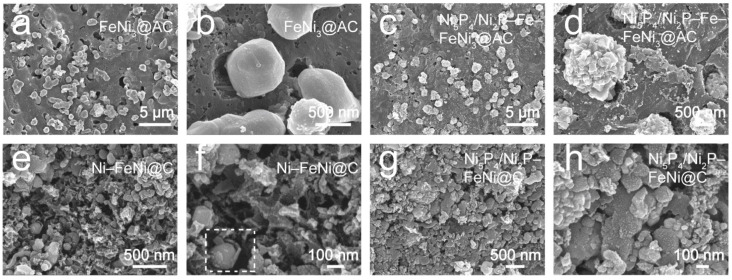
The morphology of the as-prepared samples. SEM images of FeNi_3_@AC (**a**,**b**), Ni_5_P_4_/Ni_2_P–Fe–FeNi_3_@AC (**c**,**d**), Ni–FeNi@C (**e**,**f**) and Ni_5_P_4_/Ni_2_P–FeNi@C (**g**,**h**) samples at different magnifications.

**Figure 4 nanomaterials-12-01848-f004:**
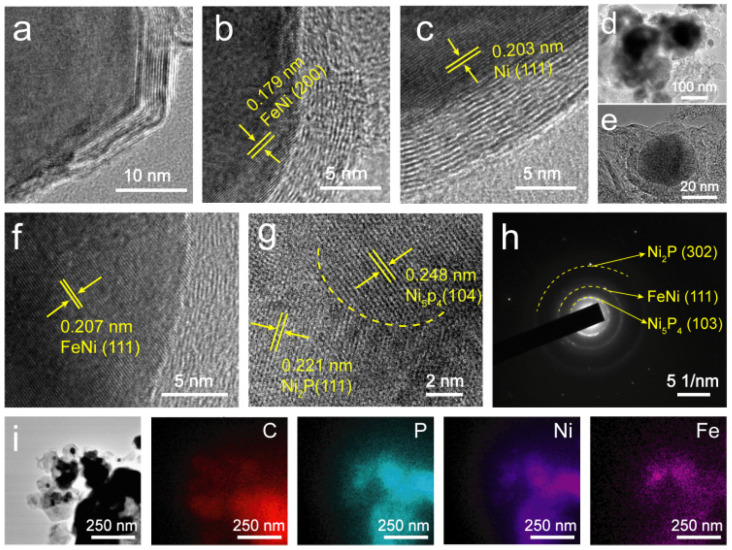
(**a**) TEM and (**b**,**c**) HRTEM images of Ni–FeNi@C. (**d**) TEM and (**e**–**g**) HRTEM images of Ni_5_P_4_/Ni_2_P–FeNi@C. (**h**) SAED and (**i**) STEM images and corresponding elemental mapping images of C, Ni, Fe and P for the Ni_5_P_4_/Ni_2_P–FeNi@C sample.

**Figure 5 nanomaterials-12-01848-f005:**
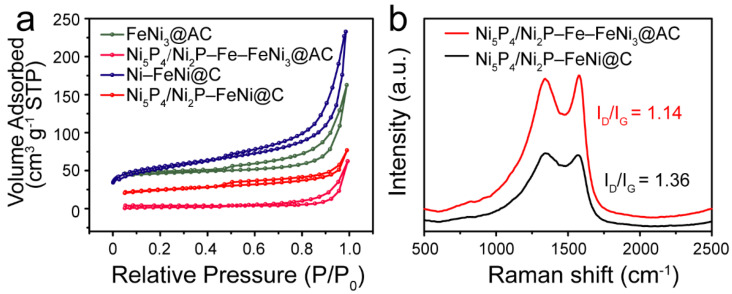
(**a**) N_2_ adsorption–desorption isotherm of all samples; (**b**) Raman spectrum of Ni_5_P_4_/Ni_2_P–Fe–FeNi_3_@AC and Ni_5_P_4_/Ni_2_P–FeNi@C.

**Figure 6 nanomaterials-12-01848-f006:**
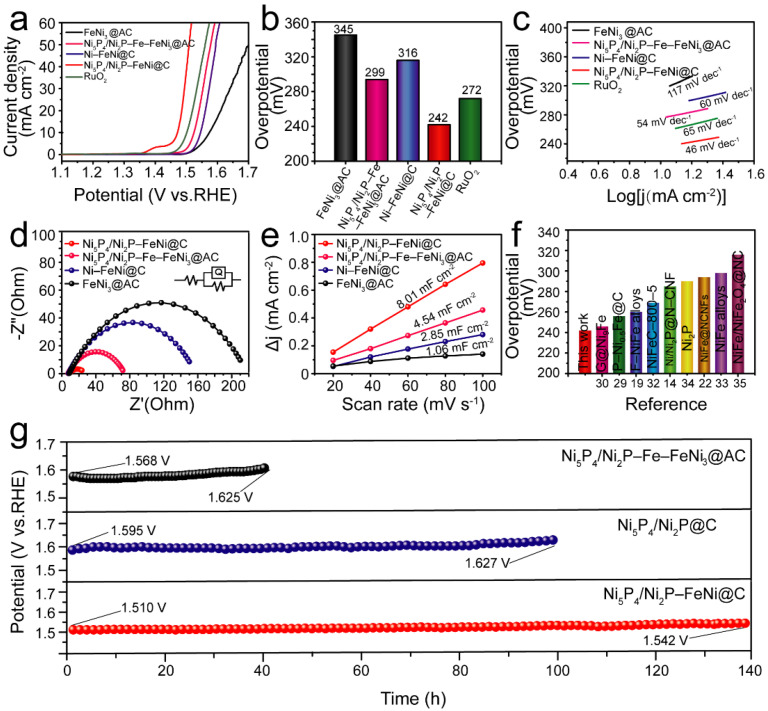
(**a**) OER LSV curves of FeNi_3_@AC, Ni–FeNi@C, Ni_5_P_4_/Ni_2_P–Fe–FeNi_3_@AC, Ni_5_P_4_/Ni_2_P–FeNi@C and the commercial RuO_2_ samples with 85% iR-correction in 1 M KOH. (**b**) The comparative overpotentials of catalysts at 10 mA cm^−2^. (**c**) Tafel plots derived from the polarization curves in (**a**). (**d**) Nyquist plots of the FeNi_3_@AC, Ni–NiFe@C, Ni_5_P_4_/Ni_2_P–Fe–FeNi_3_@AC and Ni_5_P_4_/Ni_2_P–FeNi@C catalysts at 1.480 V vs. RHE. (**e**) Double-layer capacitance (*C*_dl_) plots. (**f**) Comparison of the overpotentials of Ni_5_P_4_/Ni_2_P–FeNi@C (this work) and the relevant OER electrocatalysts at 10 mA cm^−2^. (**g**) Chronopotentiometry curves of the Ni_5_P_4_/Ni_2_P–FeNi@C, Ni_5_P_4_/Ni_2_P–Fe–FeNi_3_@AC and Ni_5_P_4_/Ni_2_P@C catalysts at 20 mA cm^−2^.

**Figure 7 nanomaterials-12-01848-f007:**
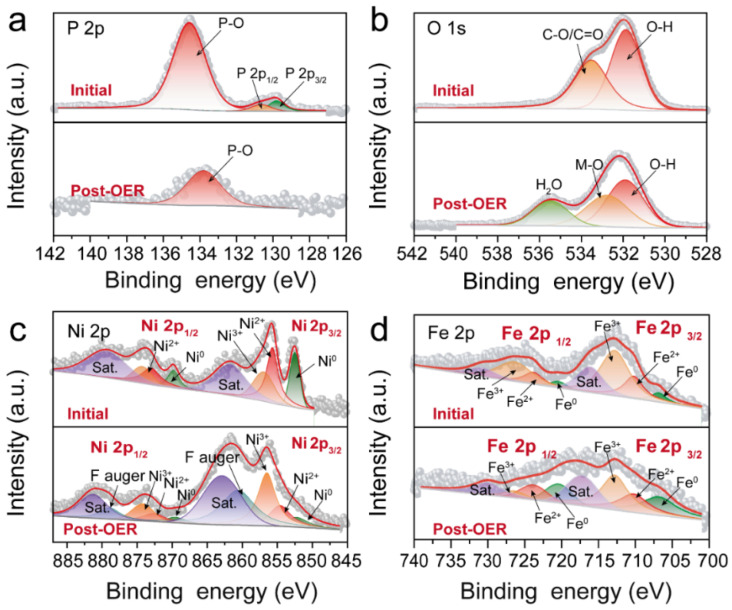
High-resolution XPS spectra of the Ni_5_P_4_/Ni_2_P–FeNi@C sample before and after the OER process. (**a**) P 2p, (**b**) O 1s, (**c**) Ni 2p and (**d**) Fe 2p.

**Figure 8 nanomaterials-12-01848-f008:**
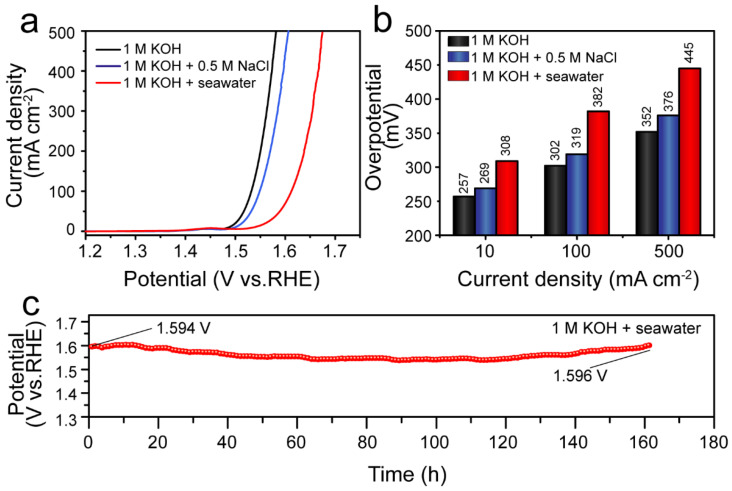
(**a**) LSV polarization curves of Ni_5_P_4_/Ni_2_P–FeNi@C with 85% iR-correction in different electrolytes. (**b**) The comparative overpotential values derived from the OER polarization curves at 20, 100 and 500 mA cm^−2^. (**c**) Chronopotentiometry curve of Ni_5_P_4_/Ni_2_P–FeNi@C at 20 mA cm^−2^ in a 1 M KOH + seawater electrolyte.

**Table 1 nanomaterials-12-01848-t001:** Comparison of OER activity of different electrocatalysts at 10 mA cm^−2^.

Catalyst	Substrate	Electrolyte	η (mV) 10 mA/cm^2^	Tafel Slope (mV/Decade)	Reference
G@Ni_9_Fe	GCE ^1^	1.0 M KOH	246	46	[30]
P–Ni_0.5_Fe@C	GCE ^1^	1.0 M KOH	256	65	[29]
F–NiFe alloys	Ni plate	1.0 M KOH	260	53	[19]
NiFeC-800-5	GCE ^1^	1.0 M KOH	269	72	[32]
Ni/Ni_2_P@N–CNF	GCE ^1^	1.0 M KOH	285	45.2	[14]
Ni_2_P	NF ^2^	1.0 M KOH	290	47	[34]
NiFe@NCNFs	GCE ^1^	1.0 M KOH	294	52	[22]
NiFe alloys	GCE ^1^	1.0 M KOH	298	51.9	[33]
NiFe/NiFe_2_O_4_@NC	GCE ^1^	1.0 M KOH	316	60	[35]
Ni_5_P_4_/Ni_2_P–FeNi@C	GCE ^1^	1.0 M KOH	242	46	This work
Commercial RuO_2_	GCE ^1^	1.0 M KOH	272	65	This work

^1^ GCE stands for glassy carbon electrode; ^2^ NF stands for nickel foam.

## Data Availability

The data that support the findings of this study are available from the corresponding authors upon reasonable request.

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
