# Peer review of "The Scalable Solid-State Synthesis of a Ni5P4/Ni2P–FeNi Alloy Encapsulated into a Hierarchical Porous Carbon Framework for Efficient Oxygen Evolution Reactions"

_nanomaterials, 2022, doi:10.3390/nano12111848_

Round 1

Reviewer 1 Report

High-performance and cheap electrocatalysts for oxygen evolution reaction  (OER) are needed, especially in large-scale water electrolysis. The authors of the publication developed a new material: three-dimensional, hierarchical, porous, based on a graphite structure (Ni5P4/Ni2P-FeNi@C). The material has a large surface area with a large number of active sites and a hierarchical porous structure, and efficiently transports electrons and ensures efficient mass diffusion. The graphite layer of the carbon coating inhibits the corrosion of the alloy particles, which results in its long-term catalytic durability. This material exhibits remarkable OER activity with low overpotential.

A comparison of the various materials described in the literature with that shown in this publication is found in Table S1 in Supporting Information. It would be good to have this Table S1 in the main article. Also, please add information about RuO2 to the table.

I recommend this article to be published in Nanomaterials with minor corrections.

Author Response

Thank you very much for the recommendation and the following invaluable comments. According to the following specific comments, we have carefully revised and improved the manuscript to merit its publication in Nanomaterials.

As you suggested, we have revised Table S1 and placed it in the main article (Table 1), details are available in the manuscript.

Table 1. Comparison of OER activity of different electrocatalysts at 10 mA cm-2.

Catalysts

Substrate

Electrolyte

η (mV)

10 mA/cm2

Tafel slope
mV/decade

Reference

G@Ni9Fe

GCE1

1.0 M KOH

246

46

30

P-Ni0.5Fe@C

GCE1

1.0 M KOH

256

65

29

F-NiFe alloys

Ni plate

1.0 M KOH

260

53

19

NiFeC-800-5

GCE1

1.0 M KOH

269

72

32

Ni/Ni2P@N-CNF

GCE1

1.0 M KOH

285

45.2

14

Ni2P

NF2

1.0 M KOH

290

47

34

NiFe@NCNFs

GCE1

1.0 M KOH

294

52

22

NiFe alloys

GCE1

1.0 M KOH

298

51.9

33

NiFe/NiFe2O4@NC

GCE1

1.0 M KOH

316

60

35

Ni5P4/Ni2P-FeNi@C

GCE1

1.0 M KOH

242

46

This work

Commercial RuO2

GCE1

1.0 M KOH

272

65

This work

1 GCE stands for the glassy carbon electrode;

2 NF stands for the nickel foam;

Reviewer 2 Report

The description of the scheme 1 needs to be improved, by using the same labels of the figure

The description of Figure 1 is missing. There are the results, but the XRD patterns are not described

It would be appropriate to add the name of the electrodes in the SEM images of Figure 2, to better understand the results

Line 270: the less BET surface areas of the phosphatized products need to be explained more in depth, since this result is counterintuitive  

Line 298: The name of the best electrode is missing in the description

Line 326: More details about the ECSA determination are required. The references are not enough

Line 392: the formation of the NiOOH seems to be crucial for high catalytic OER activity. XPS analysis of the other phosphatized product cold demonstrate that this compounds is formed only when the more performant electrode is used, by consolidating the experimental results

Reviewer 3 Report

The authors developed an in-situ generation of the template and pore-former strategy to encapsulate Ni5P4/Ni2P heterojunction and dispersive FeNi alloy particles hybrid into 3D hierarchical porous graphitic carbon framework via a room temperature solid-state grind and sodium carbonate-assisted pyrolysis method for the OER applications. This article can be published after the following  corrections:

  1.  The author should elaborate on the novelty of this work compared to this work in the introduction sections.
  2. The plane related to every peak at the Fig. 1 XRD should be indicated from their respective JCPDS values.
  3. The elemental ratio and the EDS spectra related to Fig. 3 (i) need to add for the complete visualization of all the elements present in the final electrode materials.
  4. The pore size distribution  (PSD) curves related to  Fig. 4 (a) need to be added with the average pore size and pore volume ratio of the as-prepared electrocatalyst materials. (Only 2 sample PSD curves is added to the Fig. S6)
  5.  Some similar types of work should be taken as the reference for the data presentation and explanation with the proper citation in the introduction section: Materials Today Nano,Vol. 17, March 2022, 100146 and Journal of Colloid and Interface Science,Vol. 618, 15 July 2022, Pages 475-482.
  6. Authors should clearly explain the iR correction during the overpotential calculation.

Round 2

Reviewer 2 Report

Accepted in present form

Reviewer 3 Report

All the comments are addressed professionally.